# Contextual Convolutional Networks

**Shuxian Liang[1,2,*], Xu Shen[2], Tongliang Liu[3], Xian-Sheng Hua[1,†]**

[1]Zhejiang University,   [2]Alibaba Cloud Computing Ltd.,
[3]Sydney AI Centre, The University of Sydney

shuxian.lsx@zju.edu.cn, shenxuustc@gmail.com,
tongliang.liu@sydney.edu.au, huaxiansheng@gmail.com

## Abstract

This paper presents a new Convolutional Neural Network, named Contextual Convolutional Network, that capably serves as a general-purpose backbone for visual recognition. Most existing convolutional backbones follow the representation-to-classification paradigm, where representations of the input are firstly generated by category-agnostic convolutional operations, and then fed into classifiers for specific perceptual tasks (e.g., classification and segmentation). In this paper, we deviate from this classic paradigm and propose to augment potential category memberships as contextual priors in the convolution for contextualized representation learning. Specifically, top-$k$ likely classes from the preceding stage are encoded as a contextual prior vector. Based on this vector and the preceding features, offsets for spatial sampling locations and kernel weights are generated to modulate the convolution operations. The new convolutions can readily replace their plain counterparts in existing CNNs and can be easily trained end-to-end by standard back-propagation without additional supervision. The qualities of Contextual Convolutional Networks make it compatible with a broad range of vision tasks and boost the state-of-the-art architecture ConvNeXt-Tiny by $1.8\%$ on top-1 accuracy of ImageNet classification. The superiority of the proposed model reveals the potential of contextualized representation learning for vision tasks. Code is available at: https://github.com/liang4sx/contextual_cnn.

## 1 Introduction

Beginning with the AlexNet (Krizhevsky et al., 2012) and its revolutionary performance on the ImageNet image classification challenge, convolutional neural networks (CNNs) have achieved significant success for visual recognition tasks, such as image classification (Deng et al., 2009), instance segmentation (Zhou et al., 2017) and object detection (Lin et al., 2014). Lots of powerful CNN backbones are proposed to improve the performances, including greater scale (Simonyan & Zisserman, 2014; Szegedy et al., 2015; He et al., 2016), more extensive connections (Huang et al., 2017; Xie et al., 2017; Sun et al., 2019; Yang et al., 2018), and more sophisticated forms of convolution (Dai et al., 2017; Zhu et al., 2019; Yang et al., 2019). Most of these architectures follow the representation-to-classification paradigm, where representations of the input are firstly generated by category-agnostic convolutional operations, and then fed into classifiers for specific perceptual tasks. Consequently, all inputs are processed by consecutive static convolutional operations and expressed as universal representations.

In parallel, in the neuroscience community, evidence accumulates that human visual system integrates both bottom-up processing from the retina and top-down modulation from higher-order cortical areas (Rao & Ballard, 1999; Lee & Mumford, 2003; Friston, 2005). On the one hand, the bottom-up processing is based on feedforward connections along a hierarchy that represents progressively more complex aspects of visual scenes (Gilbert & Sigman, 2007). This property is shared with the aforementioned representation-to-classification paradigm (Zeiler & Fergus, 2014; Yamins et al., 2014). On the other hand, recent findings suggest that the top-down modulation affects the bottom-up processing in a way that enables the neurons to carry more information about the stimulus being

---

[*]This work was done when the author was visiting Alibaba as a research intern.

[†]Corresponding author.

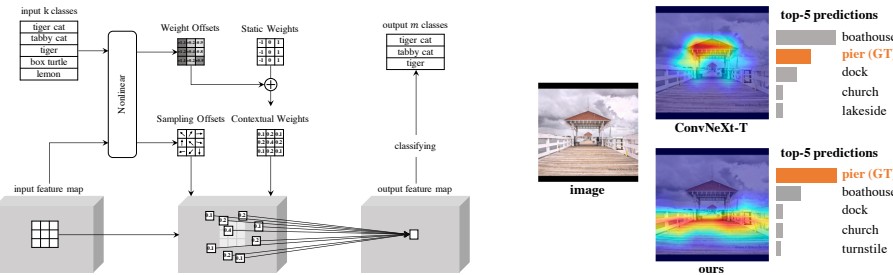

(a) $3 \times 3$ contextual convolution       (b) visualization of learned features

Figure 1: **Left**: Illustration of a $3 \times 3$ contextual convolution. Given preceding instance features and top-$k$ likely classes, sampling offsets and weight offsets are generated via non-linear layers. These offsets are added to regular grid sampling locations and static kernel weights of a standard convolution, respectively. **Right**: Grad-CAM visualization (Selvaraju et al., 2017) of the learned features of ConvNeXt-T (Liu et al., 2022) and our model. Grad-CAM is used to interpret the learned features by highlighting corresponding regions that discriminate the predicted class from other classes.

discriminated (Gilbert & Li, 2013). For example, recordings in the prefrontal cortex reveal that the same neuron can be modulated to express different categorical representations as the categorical context changes (e.g., from discriminating animals to discriminating cars) (Cromer et al., 2010; Gilbert & Li, 2013). Moreover, words with categorical labels (e.g., "chair") can set visual priors that alter how visual information is processed from the very beginning, allowing for more effective representational separation of category members and nonmembers (Lupyan & Ward, 2013; Boutonnet & Lupyan, 2015). The top-down modulation can help to resolve challenging vision tasks with complex scenes or visual distractors. This property is however not considered by recent CNN backbones yet.

Motivated by the top-down modulation with categorical context in the brain, we present a novel architecture, namely Contextual Convolutional Networks (Contextual CNN), which augments potential category memberships as contextual priors in the convolution for representation learning. Specifically, the top-$k$ likely classes by far are encoded as a contextual vector. Based on this vector and preceding features, offsets for spatial sampling locations and kernel weights are generated to modulate the convolutional operations in the current stage (illustrated in Fig. 1a). The sampling offsets enable free form deformation of the local sampling grid considering the likely classes and the input instance, which modulates *where* to locate information about the image being discriminated. The weight offsets allow the adjustment of specific convolutional kernels (e.g. "edges" to "corners"), which modulates *how* to extract discriminative features from the input image. Meanwhile, the considered classes are reduced from $k$ to $m$ ($m < k$) and fed to the following stage for further discrimination. By doing so, the following stage of convolutions is conditioned on the results of the previous, thus rendering convolutions dynamic in a smart way.

The proposed contextual convolution can be used as a drop-in replacement for existing convolutions in CNNs and trained end-to-end without additional supervision. Serving as a general-purposed backbone, the newly proposed Contextual CNN is compatible with other backbones or methods in a broad range of vision tasks, including image classification, video classification and instance segmentation. Its performance surpasses the counterpart models by a margin of $+1.8\%$ top-1 accuracy (with $3\%$ additional computational cost) on ImageNet-1K (Deng et al., 2009), $+2.3\%$ top-1 accuracy on Kinetics-400 (Kay et al., 2017), $+1.1\%$ box AP and $+1.0\%$ mask AP on COCO (Lin et al., 2014), demonstrating the potential of contextualized representation learning for vision tasks. The qualitative results also reveal that Contextual CNN can take on selectivity for discriminative features according to the categorical context, functionally analogous to the top-down modulation of the human brain. As shown in Figure 1b, the counterpart model presents high but wrong score for "boathouse" w.r.t. the groundtruth class "pier" based on features of the oceanside house, which are shared across images of both classes. In contrast, the proposed model predicts correctly by generating features of the long walkway stretching from the shore to the water, which is a crucial cue to discriminate "pier" from "boathouse". We hope that Contextual CNN's strong performance on various vision problems can promote the research on a new paradigm of convolutional backbone architectures.

## 2 RELATED WORKS

**Classic CNNs.** The exploration of CNN architectures has been an active research area. VGG nets (Simonyan & Zisserman, 2014) and GoogLeNet (Szegedy et al., 2015) demonstrate the benefits of increasing depth. ResNets (He et al., 2016) verify the effectiveness of learning deeper networks via residual mapping. Highway Network adopts a gating mechanism to adjust the routing of short connections between layers. More recently, some works include more extensive connections to further improve the capacity of CNNs. For example, DenseNet (Huang et al., 2017) connects each layer to every other layer. ResNeXt (Xie et al., 2017) aggregates a set of transformations via grouped convolutions. SENet (Hu et al., 2018) recalibrates channel-wise feature response using global information. HRNet (Wang et al., 2020) connects the high-to-low resolution convolution streams in parallel. FlexConv (Romero et al., 2022) learns the sizes of convolutions from training data. Other recent works improve the efficiency of CNNs by introducing depthwise separable convolutions (Howard et al., 2017) and shift operation (Wu et al., 2018a).

**Dynamic CNNs.** Different from the classic CNNs, dynamic CNNs adapt their structures or parameters to the input during inference, showing better accuracy or efficiency for visual recognition. One line of work drops part of an existing model based on the input instance. For example, some works skip convolutional layers on a per-input basis, based on either reinforcement learning (Wang et al., 2018; Wu et al., 2018b) or early-exit mechanism (Huang et al., 2018). The other line of work uses dynamic kernel weights or dynamic sampling offsets for different inputs. Specifically, some works aggregate multiple convolution kernels using attention (Yang et al., 2019; Chen et al., 2020) or channel fusion (Li et al., 2021). WeightNet Ma et al. (2020) unifies kernel aggregation and the channel excitation via grouped fully-connected layers. Dai et al. (2017) and Zhu et al. (2019) learn different sampling offsets of convolution for each input image. The proposed Contextual CNN shares some similar high level spirit with these dynamic CNNs. A key difference of our method is that we explicitly adopt potential category memberships as contextual priors to constrain the adaptive inference.

There are some recent CNN architectures which have shared the same term "context" (Duta et al., 2021; Marwood & Baluja, 2021). The differences of our method lie in two folds. First, the context in our work refers to the category priors (i.e., top-k likely classes) of each input while theirs refer to the boarder receptive field of the convolutions. Second, our work modulates convolutional kernels (via weight/sampling offsets) according to the category priors. They adopt multi-level dilated convolutions and use soft attention on spatial & channel dimensions, respectively. Neither of them leverages the category priors, nor modulates the convolutions to extract category-specific features.

## 3 CONTEXTUAL CONVOLUTIONAL NETWORKS

In this section, we describe the proposed Contextual Convolutional Networks. The overall architecture is introduced in Section 3.1. In Section 3.2, we present contextualizing layers, which generate contextual priors for Contextual CNN. In Section 3.3, we present contextual convolution blocks, which modulate convolutions according to the contextual priors. Without loss of generality, this section is based on the ConvNeXt-T version (Liu et al., 2022), namely Contextual ConvNeXt-T. Detailed architecture specifications of Contextual CNN are in the supplementary material.

### 3.1 OVERALL ARCHITECTURE

An overview of Contextual CNN is presented in Figure 2. Consider an input RGB image of size $H \times W \times 3$, where $H$ is the height and $W$ is the width. $N_1$ is the number of all possible classes for the image (for example, $N_1$ is 1000 for ImageNet-1K (Deng et al., 2009)). A set of class embeddings $\mathcal{E}_1 = \left\{ e_1^1, e_1^2, \cdots, e_1^{N_1} \right\}, e_1^i \in \mathbb{R}^d$ is generated from an embedding layer of size $N_1 \times d$. These class embeddings are constant in the whole network and form the basis of contextual priors.

Following common practice in prior vision backbones (He et al., 2016; Xie et al., 2017; Radosavovic et al., 2020; Liu et al., 2022), Contextual CNN consists of a stem ("*Stem*") that preprocesses the input image, a network body ("*Stage1-4*") that generates the feature maps of various granularities and a final head that predicts the output class. These components are introduced as follows.

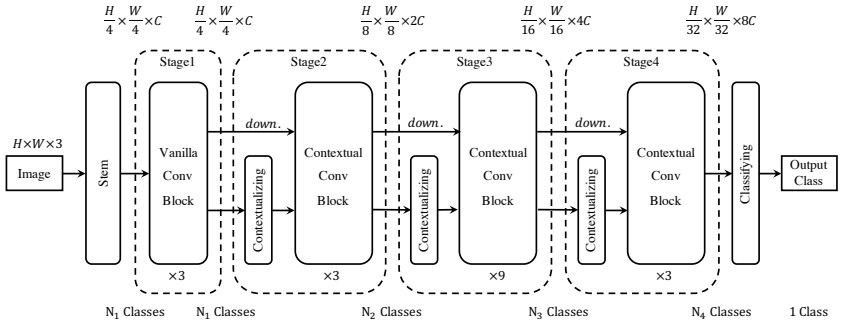

Figure 2: The architecture of a Contextual Convolutional Network (Contextual ConvNext-T). For simplicity, we denote the downsampling layers at "*Stage2-4*" by "*down.*".

**The stem.** Consistent with the standard design of ConvNeXt-T, the stem of Contexual ConvNeXt-T uses a $4 \times 4$, stride 4 convolution. The stem results in a $4\times$ downsampling of the input image, while the output features have $C = 96$ channels.

**The body.** Several vanilla convolutional blocks are applied on the output features of the stem. Maintaining the resolution of $\frac{H}{4} \times \frac{W}{4}$, these blocks have an output channel size of $C$ and share the same architecture as their ConvNeXt counterparts. These blocks are referred to as "*Stage1*" in this work. A **contextualizing layer** (described in Section 3.2) is used afterwards to extract the contextual prior from the most likely classes. It reduces the number of considered classes from $N_1$ to $N_2$ and merges the embeddings of the output $N_2$ classes as a contextual prior. In parallel, a vanilla convolutional layer is used to $2\times$ downsample the resolution to $\frac{H}{8} \times \frac{W}{8}$ and doubles the channel size to $2C$. Taking the contextual prior and the downsampled feature map as inputs, a few **contextual convolution blocks** (described in Section 3.3) are applied for feature extraction. The contextualizing layer, the downsampling layer and the following blocks are denoted as "*Stage 2*". The procedure is repeated twice, in "*Stage 3*" and "*Stage 4*". Noting that "*Stage 3*" has a feature map resolution of $\frac{H}{16} \times \frac{W}{16}$, maintains $4C$ channels and reduces the number of considered classes to $N_3$. "*Stage 4*" has a feature map resolution of $\frac{H}{32} \times \frac{W}{32}$, maintains $8C$ channels and reduces the number of considered classes to $N_4$. As the number of considered classes reduces gradually, the contextual prior conveys more fine-grained categorical information (shown in Fig. 1a), which modulates the higher-level contextual convolutions to extract more discriminative features accordingly. Details of choosing the numbers of considered classes ($N_2$, $N_3$ and $N_4$) are in the supplementary material.

**The head**. The head of Contextual CNN shares the same procedure of reducing considered classes in the contextualizing layers (i.e., classifying in Section 3.2). And it finally reduces the number of classes from $N_4$ to 1. The final one class is used as the output class for the input image.

**The loss**. The number of considered classes are reduced progressively in Contextual CNN ($N_1 \rightarrow N_2 \rightarrow N_3 \rightarrow N_4 \rightarrow 1$). For stage $i$ ($i \in \{1, 2, 3, 4\}$), we adopt a cross entropy loss over the corresponding classification scores $s_i$ (introduced in Section 3.2). Following Radford et al. (2021), given the set of considered classes $\mathcal{N}_i$ ($|\mathcal{N}_i| = N_i$), the loss is calculated by:

$$\mathcal{L}_i = -\mathbb{I}\left(y \in \mathcal{N}_i\right) \cdot \log \frac{exp\left(s_i\left(y\right)/\tau\right)}{\sum_{j=1}^{N_i} \exp\left(s_i\left(\mathcal{N}_i^j\right)/\tau\right)}, \tag{1}$$

where $\mathbb{I}$ is an indicator function, $y$ denotes the ground-truth class and $\tau$ is a learnable temperature parameter. The overall loss of Contextual CNN is computed by:

$$\mathcal{L} = \alpha\left(\mathcal{L}_1 + \mathcal{L}_2 + \mathcal{L}_3\right) + \mathcal{L}_4, \tag{2}$$

where $\alpha$ is a weight scalar. It is empirically set as $0.15$ for all experiments in this paper. More discussions about $\alpha$ are in the supplementary material.

## 3.2 CONTEXTUALIZING LAYERS

Each contextualizing layer consists of two steps. The first step, dubbed classifying, reduces the number of considered classes from $N_i$ to $N_{i+1}$, where $i$ is the index of the corresponding stage. The

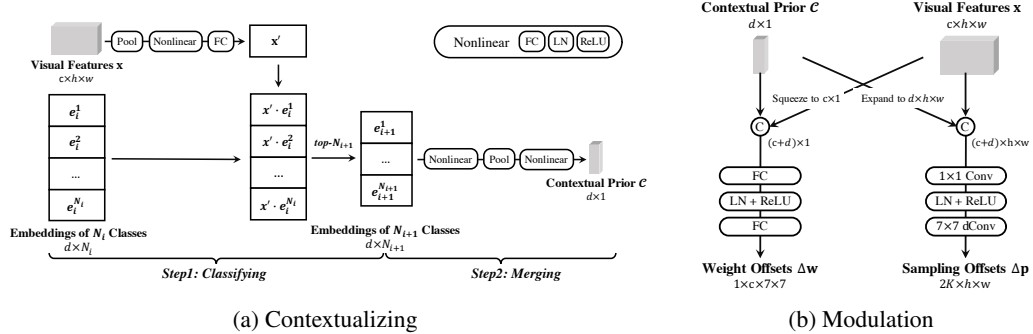

(a) Contextualizing

(b) Modulation

Figure 3: Illustration of (a) contextualizing and (b) modulation using contextual prior and features.

second step, dubbed merging, merges the embeddings of $N_{i+1}$ considered classes into a contextual prior for later contextual convolution blocks.

**Classifying**. As shown in Figure 3a, given the $N_i$ considered classes, a set of class embeddings $\mathcal{E}_i = \left\{ e_i^1, e_i^2, \cdots, e_i^{N_i} \right\}$ is collected from the aforementioned embedding layer. Following Radford et al. (2021), to compare with these embeddings, the visual features $\mathbf{x}$ from the proceding stage are projected to the embedding space. The projection involves a global average pooling and two fully-connected (FC) layers. Layer Normalization (LN) (Ba et al., 2016) and ReLU (Nair & Hinton, 2010) are used between the FC layers. The output of the projection is a visual feature vector with dimension $d$. Then, cosine similarity is computed between the L2-normalized visual feature vector and the L2-normalized embeddings of the $N_i$ classes. The resulting similarity vector $s_i$ is used as the classification scores of the $N_i$ classes for loss calculation. The top-$N_{i+1}$ highest scoring classes in $s_i$ are collected and propagated to the following merging step as well as the next stage.

**Merging**. Given $N_{i+1}$ output classes from the classifying step, we merge their embeddings $\mathcal{E}_{i+1} = \left\{ e_{i+1}^1, e_{i+1}^2, \cdots, e_{i+1}^{N_{i+1}} \right\}$ into the contextual prior $\mathcal{C} \in \mathbb{R}^{d \times 1}$. Specifically, the merging operation uses two fully connected layers (followed by LN and ReLU) and a 1D global average pooling layer between them. The merging layers are different between stages. The generated context prior $\mathcal{C}$ summarizes the similarities and differences between the considered classes. It acts as task information for the extraction of more discriminative features in the following contextual convolution blocks.

### 3.3 CONTEXTUAL CONVOLUTION BLOCK

A vanilla ConvNeXt block contains one $7 \times 7$ depthwise convolution and two $1 \times 1$ pointwise convolutions. A contextual convolution block in Contextual ConvNeXt-T is built by replacing the $7 \times 7$ depthwise convolution by a $7 \times 7$ contextual convolution with other layers unchanged.

For a vanilla $7 \times 7$ depthwise convolution, consider the convolution kernel $\mathbf{w}$ of size $1 \times c \times 7 \times 7$, where $c$ is the input channel size[1]. For each position $\mathbf{p}$ on the output feature map $\mathbf{y}$, the convolution first samples $K = 49$ locations over the input feature map $\mathbf{x}$ using a regular grid $\mathbf{g}$, then sums all sampled values weighted by $\mathbf{w}$:

$$\mathbf{y}(\mathbf{p}) = \sum_{k=1}^{K} \mathbf{w}(k) \cdot \mathbf{x}(\mathbf{p} + \mathbf{g}(k)). \tag{3}$$

In contextual convolutions, we augment the kernel weight $\mathbf{w}$ with weight offsets $\Delta\mathbf{w}$, and augment the grid $\mathbf{g}$ with sampling offsets $\Delta\mathbf{p}$:

$$\mathbf{y}(\mathbf{p}) = \sum_{k=1}^{K} \left( \mathbf{w}(k) + \Delta\mathbf{w}(k) \right) \cdot \mathbf{x}(\mathbf{p} + \mathbf{g}(k) + \Delta\mathbf{p}(k)). \tag{4}$$

**Weight offsets.** The weight offsets $\Delta\mathbf{w}$ allow the adaptive adjustment of the convolution weights according to the contextual priors. As illustrated in Figure 3b, to obtain $\Delta\mathbf{w}$, we squeeze the input

---

[1]Depthwise convolutions operate on a per-channel basis and do not change the channel size of features.

map $\mathbf{x}$ via global average pooling (GAP) and then concatenate the resulting feature vector with the contextual prior $\mathcal{C}$. Two FC layers with LN and ReLU between them are applied afterwards to generate $\Delta \mathbf{w}$. Notably, we configure the size of $\Delta \mathbf{w}$ as $1 \times c \times 7 \times 7$, same as the dimensions of $w$, to allow the summation in equation 4.

**Sampling offsets.** Following Dai et al. (2017) and Zhu et al. (2019), the sampling offsets $\Delta \mathbf{p}$ are applied to enable free-form deformation of the sampling grid. In our case, we compute the sampling offsets considering not only the input features but also the contextual priors. As shown in Figure 3b, inspired by Liang et al. (2022), we first expand the contextual prior $\mathcal{C}$ to the same spatial shape of $\mathbf{x}$, then concatenate them along the channel dimension. The resulting maps are then fed to a nonlinear convolutional block consisting of one $1 \times 1$ convolution (followed by LN and ReLU) and one $7 \times 7$ depthwise convolution (with the same kernel size and dilation as the replaced vanilla convolution). The output sampling offsets have the same resolution as the input features. The channel size $2K$ corresponds to $K$ 2D offsets.

To balance accuracy and efficiency, only one of every three blocks is replaced by contextual convolution block in Contextual ConvNeXt-T. More details are in the supplementary material.

## 4 EXPERIMENTS

In the following, Contextual CNN is compared with the state of the arts (SOTAs) on three tasks, i.e., image classification, video classification and instance segmentation. We then ablate important design elements and analyze internal properties of the method via exemplification and visualizations.

### 4.1 IMAGE CLASSIFICATION ON IMAGENET-1K

**Settings**. For image classification, we benchmark Contextual CNN on ImageNet-1K (Deng et al., 2009). It contains 1.28M training images and 50K validation images from $1,000$ classes. The top-1 accuracy on a single crop of size $224 \times 224$ is reported. To compare with SOTAs, we instantiate Contextual CNN using the recent method ConvNeXt (Liu et al., 2022), dubbed Contextual ConvNeXt. Following Touvron et al. (2021); Liu et al. (2021; 2022), we train the model for 300 epochs using an AdamW optimizer (Loshchilov & Hutter, 2017) with a learning rate of 0.001. The batch size we use is $4,096$ and the weight decay is 0.05. We adopt the same augmentation and regularization strategies as Liu et al. (2022) in training. Unless otherwise specified, the channel size of class embeddings $d$ is 256. The numbers of considered classes for the four stages are: $N_1 = 1000$, $N_2 = 500$, $N_3 = 200$ and $N_4 = 50$. More details and discussions are in the supplementary material.

Table 1: Comparison with the state-of-the-arts on ImageNet-1K. "FLOPs" denotes multiply-add operations. Following Liu et al. (2021), inference throughput is measured on a V100 GPU.

| model | image size | #param. | FLOPs | throughput (images/s) | top-1 acc. |
|---|---|---|---|---|---|
| RegNetY-4G (Radosavovic et al., 2020) | $224^2$ | 21M | 4.0G | 1156.7 | 80.0 |
| RegNetY-8G (Radosavovic et al., 2020) | $224^2$ | 39M | 8.0G | 591.7 | 81.7 |
| RegNetY-16G (Radosavovic et al., 2020) | $224^2$ | 84M | 16.0G | 334.7 | 82.9 |
| Swin-T (Liu et al., 2021) | $224^2$ | 28M | 4.5G | 757.9 | 81.3 |
| Swin-S (Liu et al., 2021) | $224^2$ | 50M | 8.7G | 436.7 | 83.0 |
| Swin-B (Liu et al., 2021) | $224^2$ | 88M | 15.4G | 286.6 | 83.5 |
| ConvNeXt-T (Liu et al., 2022) | $224^2$ | 29M | 4.5G | 774.7 | 82.1 |
| ConvNeXt-S (Liu et al., 2022) | $224^2$ | 50M | 8.7G | 447.1 | 83.1 |
| ConvNeXt-B (Liu et al., 2022) | $224^2$ | 89M | 15.4G | 292.1 | 83.8 |
| Contextual ConvNeXt-T (ours) | $224^2$ | 32M | 4.6G | 770.6 | **83.9** |
| Contextual ConvNeXt-S (ours) | $224^2$ | 53M | 8.9G | 445.2 | **84.6** |
| Contextual ConvNeXt-B (ours) | $224^2$ | 93M | 15.8G | 291.0 | **85.2** |

**Results**. Table 1 presents the ImageNet-1K results of various SOTA architectures, including Reg-Net (Radosavovic et al., 2020), Swin Transformer (Liu et al., 2021) and ConvNeXt (Liu et al., 2022). Contextual ConvNeXt outperforms all these architectures with similar complexities, e.g., +1.8%/+1.5%/+1.4% vs. ConvNeXt-T/S/B. The above results verify the effectiveness of Contextual CNN for large-scale image classification, showing the potential of contextualized representation learning. Inspired by Swin Transformer, to compare efficiency with hardware-optimized classic CNNs, we adopt an efficient batch computation approach for contextual convolutions (detailed in §3 of the supplementary material). Thus, in addition to the noticeably better performances, Contextual ConvNeXt also enjoys high inference throughput comparable to ConvNeXt.

## 4.2 Empirical Evaluation on Downstream tasks

Table 2: Kinetics-400 video action classifcation results using TSM (Lin et al., 2019). † denotes the reproduced results using the same training recipe.

| backbone | image size | #frames | FLOPs | top-1 | top-5 |
|---|---|---|---|---|---|
| ResNet50 | $224^2$ | 16 | 65G | 74.7 | 91.4 |
| ResNet101 † | $224^2$ | 16 | 125G | 75.9 | 92.1 |
| Contextual ResNet50 (ours) | $224^2$ | 16 | 68G | **77.0** | **93.1** |

Table 3: COCO object detection and segmentation results using Mask-RCNN. Following Liu et al. (2021), FLOPs are calculated with image size $1280 \times 800$.

| backbone | FLOPs | $AP^{box}$ | $AP^{box}_{50}$ | $AP^{box}_{75}$ | $AP^{mask}$ | $AP^{mask}_{50}$ | $AP^{mask}_{75}$ |
|---|---|---|---|---|---|---|---|
| Swin-T (Liu et al., 2021) | 267G | 46.0 | 68.1 | 50.3 | 41.6 | 65.1 | 44.9 |
| ConvNeXt-T (Liu et al., 2022) | 262G | 46.2 | 67.9 | 50.8 | 41.7 | 65.0 | 44.9 |
| Contextual ConvNeXt-T (ours) | 267G | **47.3** | **69.0** | **52.2** | **42.7** | **66.2** | **45.6** |

**Video classification on Kinetics-400**. Kinetics-400 (Kay et al., 2017) is a large-scale video action classification dataset with 240K training and 20K validation videos from 400 classes. We fine-tune a TSM (Lin et al., 2019) model with a Contextual CNN based on ResNet50 (He et al., 2016). For a fair comparison, the model is first pretrained on ImageNet-1K following the original ResNet50, then fine-tuned on Kinetics-400 following TSM. More details are in the supplementary material.

Table 2 lists the video action classification results of Contextual ResNet50 and ResNet50 / ResNet101. It is observed that Contextual ResNet50 is +2.3%/+1.7% better on top-1/5 accuracy than ResNet50 with similar computation cost. Moreover, Contextual ResNet50 is even better than the more sophisticated backbone ResNet101. These results verify that the Contextual CNN architecture can be effectively extended to general visual recognition tasks like video classification.

**Instance segmentation on COCO**. The instance segmentation experiments are conducted on COCO (Lin et al., 2014), which contains 118K training and 5K validation images. Following Swin Transformer, we fine-tune Mask R-CNN (He et al., 2017) on the COCO dataset with the aforementioned Contextual ConvNeXt backbones. The training details are deferred to the supplementary material.

Table 3 shows the instance segmentation results of Swin Transformer, ConvNeXt and Contextual ConvNeXt. With similar complexity, our method achieves better performance than ConvNeXt and Swin-Transformer in terms of both box and mask AP. This demonstrates that the superiority of Contextual CNN's contextualized representations still hold for downstream dense vision tasks, indicating that Contextual CNN is able to serve as a general-purpose backbone for computer vision.

## 4.3 Ablation Study

We ablate major design elements in Contextual CNN using ImageNet-1K dataset. All ablation models are based on ResNet50 (He et al., 2016). Details of the architecture are in the supplementary material.

**Block design (Table 4)**. We empirically analyze the effect of the proposed components in our work: contextualizing layers and contextual convolutions. First, compared with vanilla ResNet50 (*a1*), sim-

Table 4: Ablations on the proposed components. "Ctx" denotes contextualizing layers. "CtxConv" denotes contextual convolutions. "Deform" denotes deformable convolutions V2 (Zhu et al., 2019).

|  | model | #param. | FLOPs | top-1 acc. |
|---|---|---|---|---|
| *a1* | R50 | 25.5M | 4.1G | 76.58 |
| *a2* | + Ctx | 25.7M | 4.1G | 77.18 |
| *a3* | + CtxConv | 27.2M | 4.2G | 77.59 |
| *a4* | + Deform | 25.7M | 4.2G | 77.24 |
| ***a5*** | **+ Ctx + CtxConv** | **27.9M** | **4.3G** | **79.35** |

ply using contextual prior $\mathcal{C}$ as an extra input (via expanding and concatenating) to the convolutions (*a2*) shows a slightly better result (+0.60%). This reveals that feeding the contextual prior without modulations provides limited gains for representation learning. Second, applying contextual convolutions alone (*a3*) leads to a +1.01% gain on top-1 accuracy while applying deformable convolutions (Zhu et al., 2019) alone shows a +0.66% gain (*a4*). The results imply that modulating convolutions according only to input features improves the expressive power of learned representations. This *spirit* is shared between contextual convolutions, deformable convolutions and other forms of dynamic convolutions. Third, combining the two proposed components, we observe a significant gain of +2.77% compared to the classic CNN (*a5* vs. *a1*) and a gain of +1.76%/+2.11% compared to the dynamic CNNs (*a5* vs. *a3/a4*). The results verify that the categorical context provides important cues about the more discriminative directions to modulate the convolutions, highlighting the advantage of using potential category memberships as contextual priors.

Table 5: Ablations on contextual modulations.

|  | weight offset | sampling offset | #param. | FLOPs | top-1 acc. |
|---|---|---|---|---|---|
| *b1* |  |  | 25.5M | 4.1G | 77.18 |
| *b2* | ✓ |  | 27.5M | 4.2G | 78.30 |
| *b3* |  | ✓ | 25.9M | 4.3G | 78.44 |
| ***b4*** | ✓ | ✓ | **27.9M** | **4.3G** | **79.35** |

**Modulation design (Table 5).** We investigate how the two forms of contextual modulations in convolutions affect the performance. Using weight/sampling offsets alone brings in a +1.12%/+1.26% improvement (*b2/b3* vs. *b1*), which indicates that either form of the modulations leverages the categorical context to some degree. Combining the two forms leads to a more significant gain of +2.17% (*b4* vs. *b1*). This implies that the two forms of modulations complete each other.

Table 6: Ablations on contextualizing. $d$ is the dimension of class embeddings.

|  | *Stage1* | *Stage2* | *Stage3* | *Stage4* | $d$ | #param. | FLOPs | top-1 acc. |
|---|---|---|---|---|---|---|---|---|
| *c1* |  |  |  |  | 256 | 25.5M | 4.1G | 76.58 |
| *c2* |  |  |  | ✓ | 256 | 26.7M | 4.1G | 77.86 |
| *c3* |  |  | ✓ | ✓ | 256 | 27.7M | 4.2G | 78.63 |
| ***c4*** |  | ✓ | ✓ | ✓ | **256** | **27.9M** | **4.3G** | **79.35** |
| *c5* | ✓ | ✓ | ✓ | ✓ | 256 | 28.0M | 4.3G | 77.99 |
| *c6* |  | ✓ | ✓ | ✓ | 64 | 27.3M | 4.2G | 77.59 |
| *c7* |  | ✓ | ✓ | ✓ | 128 | 27.5M | 4.3G | 78.44 |
| *c8* |  | ✓ | ✓ | ✓ | 512 | 28.3M | 4.3G | 78.89 |

**Contextualizing design (Table 6).** We study the influence of two hyperparameters: the number of contextual stages and the dimension of class embeddings $d$. First, the model with 3 contextual stages ("*Stage2-4*") yields the best top-1 accuracy compared to those with more or less contextual stages (*c3* vs. *c0/c1/c2/c4*). This suggests that our contextual modulation is more effective on middle-level and high-level features. Then, by increasing the dimension of class embeddings $d$, the accuracy improves steadily and saturates at 256 (*c4* vs. *c6/c7/c8*). This reveals the strategy of setting $d$ in practice.

## 4.4 More Analysis

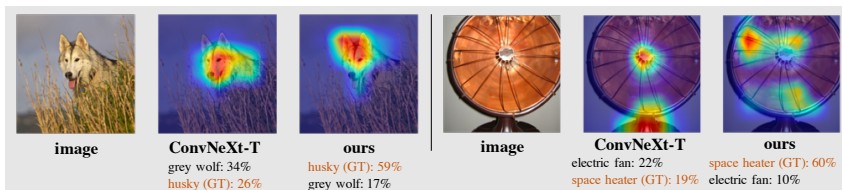

Figure 4: Comparison of Grad-CAM visualization results. The Grad-CAM visualization is calculated for the last convolutional outputs. "GT" denotes the ground truth class of the image.

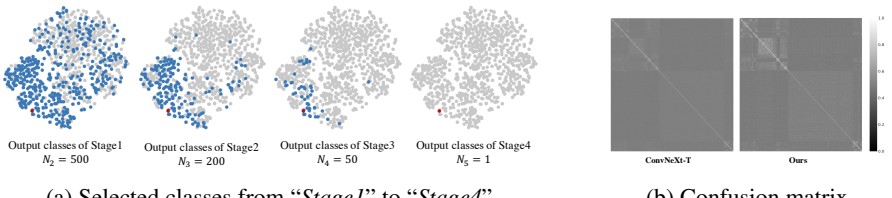

(a) Selected classes from *"Stage1"* to *"Stage4"*    (b) Confusion matrix

Figure 5: **Left**: The t-SNE distributions of the class embeddings on ImageNet-1K. The points in blue denote the selected classes of the corresponding stage and the point in red denotes the groud-truth class. **Right**: Comparison of class embeddings between ConvNeXt-T and our Contextual ConvNext-T on ImageNet-1K. For ConvNeXt-T, we normalize the weights of the classifier as its class embeddings ($1000 \times 768$). The confusion matrix denotes the similarity matrix of class embeddings ($1000 \times 1000$).

**Analysis of generated features**. We adopt Grad-CAM (Selvaraju et al., 2017) to interpret the learned features by highlighting the regions that discriminate different classes (shown in Figure 4). In the first case, the counterpart (ConvNeXt-T) generates features w.r.t. the face of the animal, which shares similar appearances between "husky" and "wolf". Our model generates features of the ears that are long and stand upright, which are the key factor to differentiate "husky" from "wolf" (offset and triangular). In the second case, the counterpart generates features w.r.t. the base and the center hub of the object, both are shared between "space heater" and "electric fan". In contrast, our model generates features of the fan region, which is very different between the two categories (filaments vs. vanes). In summary, the counterpart generates shared patterns of the most likely classes, which help select these classes out of 1,000 classes but fail to further differentiate them. This behavior is reasonable since the convolutions in the model are category-agnostic. Contextual CNN, in contrast, takes a few most likely classes as contextual priors and learns to generate more discriminative features w.r.t. these classes. Thus, our model is superior in resolving the above challenging cases that confuse the counterpart.

**Analysis of class embeddings**. Figure 5a visualizes the stage-wise classifying of Contextual CNN. Specifically, we first adopt t-SNE (Van der Maaten & Hinton, 2008) to visualize the class embeddings of the model. We then highlight the selected classes from *"Stage1"* to *"Stage4"*. The results suggest that the contextual prior progressively converges to semantic neighbors of the groundtruth class. Figure 5b compares the confusion matrix of class embeddings between ConvNext-T and Contextual ConvNeXt-T. Following Chen et al. (2019), we uses the weights of the last fully-connected layer of ConvNext-T as its class embeddings. One can observe that the class embeddings learned by our model exhibit more effective separation of category memberships.

## 5 Conclusion

This paper presents Contextual Convolutional Network, a new CNN backbone which leverages a few most likely classes as contextual information for representation learning. Contextual CNN surpasses the performance of the counterpart models on various vision tasks including image classification, video classification and object segmentation, revealing the potential of contextualized representation learning for computer vision. We hope that Contextual CNN's strong performance on various vision problems can promote the research on a new paradigm of convolutional backbone architectures.

## A    ETHICS STATEMENT

First, most modern visual models, including Swin Trasnformer, ConvNeXt and the proposed Contextual CNN, perform best with their huge model variants as well as with large-scale training. The accuracy-driven practice consumes a great amount of energy and leads to an increase in carbon emissions. One important direction is to encourage efficient methods in the field and introduce more appropriate metrics considering energy costs. Then, many of the large-scale datasets have shown biases in various ways, raising the concern of model fairness for real-world applications. While our method enjoys the merits of large-scale training, a reasonable approach to data selection is needed to avoid the potential biases of visual inference.

## B    REPRODUCIBILITY STATEMENT

We make the following efforts to ensure the reproducibility of this work. First, we present detailed architecture specifications of Contextual ConvNeXt and Contextual ResNet50 in §1 of the supplementary material. Second, we provide detailed experimental settings (e.g., the training recipes) of Contextual CNN for all involved tasks in §2 of the supplementary material. Third, code of Contextual CNN is available at: `https://github.com/liang4sx/contextual_cnn`.

## C    ACKNOWLEDGMENTS

This work was (partially) supported by the National Key R&D Program of China under Grant 2020AAA0103901.

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
