# OpenReview forum: "Contextual Convolutional Networks"
_ICLR.cc/2023/Conference — ICLR 2023 poster_

### Official Review · Reviewer_2om9 · 2022-10-21

**Confidence:** 3
**Correctness:** 4
**Technical Novelty And Significance:** 3
**Empirical Novelty And Significance:** 3
**Recommendation:** 8

**Clarity, Quality, Novelty And Reproducibility:**

The paper is very well written and easy to read and to follow.
The analysis are useful and provide the right level of illustration.

Finally a few things were not entirely clear to me
  - is the merging network the same for each stage?
  - are the class embedding constant in the whole network or are there different class embeddings after each stage?

The idea seems closely related to the topics of deformable and dynamic convolutions and, to some extent, of attention, which is clearly detailed in section 2. The main novelty lies in the conditioning of these techniques on category priors which looks interesting and novel enough.

The text looks clear enough for reproducibility, besides a few details. Reproducibility should be ensured with the availability of the code, as mentioned by the authors.

**Strength And Weaknesses:**

Strenghs:
  - The idea is simple and well-explained, and extends and combines nicely the notions of deformable convolutions and dynamic convolutions with context
  - The experiments report better results with a limited impact on computational efficiency
  - The method is simple enough and looks easy to integrate

The main weakness of the idea, compared to the conventional CNN backbones lies in the adaptation of such a model to different tasks. It might be useful to only adapt the top layer(s) for a new task. In the presented experiments, it looks like the whole network needs to be fine-tuned. One experiments is reported in which class embeddings are frozen. It would be interesting to also measure the accuracy after only fine-tuning the top layer.

**Summary Of The Paper:**

This paper addresses the problem of computer vision with convolutional neural networks (CNNs). The authors propose to contextualize the CNN by incorporating category priors. This is achieved by adding two new features to the CNN architecture.

Some convolutional layers are contextualized by deforming the convolution grid using predicted spatial offsets, and by dynamically changing the kernel weights using predicted weight offsets. The offsets are predicted from the visual features extracted at the previous layer and a context prior.

The introduced context prior is computed by a new context layer which performs two tasks. First, from the visual features are projected to the space of classes embeddings, and a cosine similarity is measured to retain the top-n classes, which will be used in the subsequent contextualized convolutions. Then, the embeddings for the retained classes are merged to compute the context prior.

Through experiments on ImageNet and two adaptation tasks, the author show that the proposed model outperforms existing methods with a limited increase of computational complexity on GPU. An ablation study shows the importance of the various features proposed.

**Summary Of The Review:**

The idea of using some context to compute the offsets of a deformable convolution is interesting and could be seen as a form of attention. For some application, it might even be more interesting to be able to select the list of classes to give to the network.

Combining this idea with the conditioning of a dynamic convolution is equally interesting.

Additionally to the GPU results, the efficiency on CPU would be nice to have, compared to an efficient convolution operation, as well as memory considerations, although this potential drawback is shared with deformable and dynamic convolutions.

Overall, I found the paper easy to read, the analysis convincing and the idea simple and interesting, worth sharing with the community.

---

> ### Author Response · Authors · 2022-11-16
> **Response to Reviewer 2om9**
>
> We sincerely thank you for taking time to carefully review our paper and for your positive feedbacks about our work. Please see our detailed responses to your comments and suggestions below.
>
> **Q1**: "It would be interesting to also measure the accuracy after only fine-tuning the top layer.".
>
> **A1**: We highly appreciate your constructive suggestion. In the table below, we report the results of only fine-tuning the top layer of Contextual ResNet50 and ResNet50 on two fine-grained downstream datasets, Stanford Cars [A] and CUB [B]. Their pretrained weights are from the ablation models $a5$ and $a1$ in Table 4, respectively. We freeze all pretrained layers (including contextualizing layers, convolutional blocks and class embeddings), remove the original final classifying head and fine-tune a randomly-initialized FC head over the backbone features.  On both datasets, the proposed method yields significantly better performance ($+5.63$%/$+3.62$%). This demonstrates that the learned features of Contextual ResNet50 are robust and can be simply transferred to fine-grained downstream tasks.
>
> |  Models  |  Cars  |  CUB  |
> | --- | --- | --- |
> | ResNet50 | 50.20% | 63.02% |
> | Contextual ResNet50 | **55.83%** | **66.64%** |
>
> **Q2**: "Are the class embedding constant in the whole network or are there different class embeddings after each stage?" and "Is the merging network the same for each stage?"
>
> **A2**: The class embeddings are **constant** in the whole network and the following merging layers are **different** for each stage. On the one hand, we have experimented with individual class embeddings for each stage and observed significant performance degradation ($-3$%) of Contextual ResNet50 on ImageNet. The reason is that, compared to the individual class embeddings, the constant class embeddings allow for more stable representations of classes throughout the whole network, and therefore ease the training of Contextual CNN. On the other hand, since different stage focuses on different aspects of discriminative features (edges/corners vs. faces/dogs) [C], the merging layers are different for each stage so that they generate the contextual priors from the corresponding aspects.
>
> **Q3**: "Additionally to the GPU results, the efficiency on CPU would be nice to have, compared to an efficient convolution operation, as well as memory considerations."
>
> **A3**: Following your valuable suggestion, we measure the CPU efficiency and the memory considerations of both ResNet50 and Contextual ResNet50. For ResNet50, the inference time on CPU is around $27$ ms/image and the memory usage is around $97$ MB. For Contextual ResNet50, the inference time on CPU is around $35$ ms/image and the memory usage is around $106$ MB. Note that the ResNet50 architecture is well optimized on both GPU and CPU. The proposed Contextual ResNet50 is optimized on GPU (detailed in $\S3$ of the supplementary material) but not on CPU. We expect that hardware-aware optimization might further improve the efficiency of the proposed method.
>
> **References**:
>
> [A] Krause, Jonathan, et al. "3d object representations for fine-grained categorization." In: ICCVW, 2013.
>
> [B] Welinder, Peter, et al. "Caltech-UCSD birds 200." (2010).
>
> [C] Zeiler, Matthew D., and Rob Fergus. "Visualizing and understanding convolutional networks." In: ECCV, 2014.

---

### Official Review · Reviewer_CrqV · 2022-10-23

**Confidence:** 4
**Correctness:** 3
**Technical Novelty And Significance:** 2
**Empirical Novelty And Significance:** 2
**Recommendation:** 6

**Clarity, Quality, Novelty And Reproducibility:**

The paper is generally clear, but I am not sure about the novelty of the method.

**Strength And Weaknesses:**

Strength:
- The paper is fairly easy to follow.
- The experiments are conducted on representative datasets.

Weaknesses:
- The authors seemed to have missed some competing works bearing almost identical names [A, B] based on the idea of leveraging the context during convolution. The authors should point the differences / novelty with respect to these existing works. To me, it seems that the proposed convolution can be seen as a generalization of the convolution proposed in [A]. I also believe a head-to-head comparison with [A, B] is in order. At the moment, I have doubts about the novelty of the method.
- An analysis of the failure cases is missing. The authors should also point out when the proposed method is expected to fail.
- Are the performance improvements statistically significant? Are the improvements consistent over multiple runs? Such questions need to be answered to validate the shown improvements.

Minor issues / language corrections:
- The citation for SEnet is missing.
- “have a output” => “have an output”.

[A] Duta et al., "Contextual Convolutional Neural Networks". In: ICCVW, 2021.
[B] Marwood et al., "Contextual Convolution Blocks". In: BMVC, 2021.

**Summary Of The Paper:**

The authors present a neural network called contextual convolutional network, which adds classification layers at multiple stages and uses the top (most probable) class embeddings in following stages. The proposed layers / blocks are added to ConvNeXt models and tested on several mainstream / large-scale datasets (ImageNet, MS COCO, Kinetics-400).

**Summary Of The Review:**

Currently, I believe the weaknesses slightly outweigh the strengths.

---

> ### Author Response · Authors · 2022-11-16
> **Response to Reviewer CrqV**
>
> All authors are grateful for your time devoted to reviewing this paper and your constructive suggestions. Please see our detailed responses to your comments and suggestions below.
>
>
> **Q1**: "The authors seemed to have missed some competing works bearing almost identical names [A, B] based on the idea of leveraging the context during convolution. The authors should point the differences / novelty with respect to these existing works. "
>
> **A1**: The differences between this work and [A, B] lie in two folds.
> - **First, the definitions of context are different.** The context in this work refers to the category priors (i.e., top-k likely classes) of each input image. The context in [A] and [B] refers to the boarder receptive field by integrating multiple levels of dilated convolutions and by extending Squeeze-and-Excitation (SE) blocks [C] with extra spatial dimensions, respectively.
> - **Second, the operations are different.** This work modulates convolutional kernels (via weight/sampling offsets) according to the category priors. This allows the convolutions to take on selectivity for more discriminative features w.r.t. a few most likely classes. [A] adopts the multi-level static convolutions on preceding features and [B] computes soft attention on spatial and channel dimensions on preceding features. Neither of them leverages the category priors or modulates the convolutions to extract category-specific features.
>
> As mentioned by Reviewer **oBzi**, this paper is the first work to reveal that conditioning convolutions on class knowledges on different stages of a CNN has a positive effect on results. As mentioned by Reviewer **2om9**, our idea is interesting and worth sharing with the community. We would be grateful if you can confirm whether our responses have addressed your concerns.
>
> **Q2**: "An analysis of the failure cases is missing. The authors should also point out when the proposed method is expected to fail."
>
> **A2**: Thanks for your constructive suggestion. The proposed method is good at handling one of the very important challenges of visual recognition, i.e., the discrimination of similar classes. However, as shown in Figure 4 of the revised supplementary material, it still fail on some other challenges, such as huge viewpoint variations (case 1 and 2), unusual object appearances (case 3 and 4) and noisy labels (case 5 and 6). Similar failure cases are also observed in recent SOTA methods like Swin Transformers and ConvNeXt. These cases reveal the unsolved challenges of visual recognition and suggest the potential directions for future works.
>
> **Q3**: "Are the performance improvements statistically significant? Are the improvements consistent over multiple runs? Such questions need to be answered to validate the shown improvements."
>
> **A3**: We highly appreciate your constructive comments. At this moment, we have trained both ResNet50 and Contextual ResNet50 ($a5$ in Table 4) on ImageNet for 5 runs. In each run, the two models share the same weight initializations (except the newly proposed layers). And the shared weight initializations are different in the $i$-th run and the $j$-th run ($i\neq j$). In this way, we obtained 5 groups of results and then conducted paired t-test between the two models. The table below shows mean accuracy and standard derivation of the models, and reports the p-value for the paired t-test. The results demonstrate that Contextual ResNet50 achieves statistically significant improvement over the baseline ResNet50 and the p-value is less than 0.05. We will add results with more runs (e.g., 15) in our camera ready version.
>
> |  Models  |  Mean $\pm$ Std. | p-value |
> | --- | --- | --- |
> | ResNet50 | 76.57 $\pm$ 0.05 | 1.19e-10 |
> | Contextual ResNet50 (ours) | 79.33 $\pm$ 0.04 | - |
>
> **Q4**: Minor issues / language corrections.
>
> **A4**: In the revised paper, we have cited and discussed the classic work SENet (Hu et al., 2018) and fixed the mentioned typo.
>
> We hope that our responses could address your concerns. Please let us know if you have further questions.
>
> **References**:
>
> [A] Duta et al., "Contextual Convolutional Neural Networks". In: ICCVW, 2021.
>
> [B] Marwood et al., "Contextual Convolution Blocks". In: BMVC, 2021.

---

> > ### Comment · Reviewer_CrqV · 2022-11-16
> > **Response to rebuttal**
> >
> > Thank you for the clarifications. Regarding Q1/A1, it would be useful to added the comments to the related work (this will also improve the related work, which is a bit short at the moment). Furthermore, to emphasize the difference and avoid confusion w.r.t. [A,B], perhaps it would be useful to disambiguate the title of the paper to something like "Category Contextual Neural Networks" or "Categorical Context Neural Networks" or "Neural Networks with Category Priors". The other points are well addressed.

---

> > > ### Author Response · Authors · 2022-11-17
> > > **Thanks for your feedback.**
> > >
> > > Thank you very much for raising the rating. We are pleased to hear that your concerns are addressed well. Following your kind suggestion, we will re-organize the related works and add the comments regarding Q1/A1 in our camera ready version. We will also highlight the definition of our context at the beginning of the paper (abstract/introduction) to avoid confusions.
> > >
> > > We sincerely thanks again for your time and efforts. Your insightful comments have helped to improve this paper a lot!

---

### Official Review · Reviewer_oBzi · 2022-10-24

**Confidence:** 4
**Correctness:** 1
**Technical Novelty And Significance:** 3
**Empirical Novelty And Significance:** 4
**Recommendation:** 8

**Clarity, Quality, Novelty And Reproducibility:**

- Clarity:
The paper is simple and clear. I do not see any issues here.

- Quality:
The quality of the results is self-explanatory based on the excellent results on ImageNet.

- Novelty:
In my opinion the contribution of this paper is very good as in my knowledge it is the first work that shows that conditioning on class knowledge on the different stages of a CNN has a positive effect on results. This shows that there is a need of more specialized layers towards the end of a CNN.

- Reproducibility:
I am happy with the level of reproducibility presented in the paper.


**Strength And Weaknesses:**

\+ The paper is well written and clear to understand
\+ Related work is complete
\+ Experimental results are thorough with the important ablation studies.

\- The convolution offsets is the same as deformable convolutions (Dai et al. 2017). How does the proposed method compares with Deformable convolutions?
\- The proposed approach has as hyperparameters the number of classes to consider on each layer. Did the authors think about considering it as a single scaling factor instead of an independent value for each stage?
\- In my understanding, in the contribution and title I would put more emphasis on the fact that the following stage of convolution depend on the results of the previous, thus rendering convolutions dynamic in a smart way.

**Summary Of The Paper:**

This paper proposes to increasingly refine at each CNN stage the knowledge about the class of a given image and use this information to build a more discriminative model. The more discriminative model is build with two contributions: i) add variations to the static CNN filters that are classes specific (based on the set of classes that are selected so far) and on the sampling location similar to deformable CNN. Both contributions seem important to boost classification results on imageNet and also other tasks,

**Summary Of The Review:**

I consider this paper a solid and balanced contribution in the field of convolutional neural networks.
It has:
- A promising contribution
- Goof and clear presentation
- Complete related work (in my knowledge)
- Excellent results & ablation studies

---

> ### Author Response · Authors · 2022-11-16
> **Response to Reviewer oBzi**
>
> All authors are grateful for your time devoted to reviewing this paper and for your positive feedbacks about our work. Please see our detailed responses to your comments and suggestions below.
>
> **Q1**: "How does the proposed method compares with Deformable convolutions? "
>
> **A1**: As shown in the table below, compared to the classic deformable convolutions V1 (Dai et al. 2017), the proposed method achieves $+2.40$% better accuracy. Compared to the latest deformable convolutions V2 (Zhu et al. 2019), the proposed method achieves $+2.11$% better accuracy. The performance gains are mainly from conditioning convolutions on category priors since the contextual convolutions w/o category priors are only slightly better than previous methods ($+0.64$%/$+0.35$%).
>
> |  Models  |  Top-1 Accuracy  |
> | --- | --- |
> | Deformable Conv V1 (Dai et al. 2017) | 76.95% |
> | Deformable Conv V2 (Zhu et al. 2019, $a4$ in Table 4) | 77.24% |
> | CtxConv w/o Category Priors ($a3$ in Table 4) | 77.59% |
> | Ours ($a5$ in Table 4) | **79.35%** |
>
> **Q2**: "Did the authors think about considering it as a single scaling factor instead of an independent value for each stage?"
>
> **A2**: In our early experiments, we have experimented with the suggested single scaling factor (e.g., $0.50$/$0.25$) and found performance degradation (e.g., $0.51$%/$0.84$%). The degradation is due to the imbalance of computational complexity between stages (e.g., 1.2M parameters in res3 and 7.1M parameters in res4). Specifically, using a large scaling factor ($0.5$), the number of considered classes reduces too slowly to leverage the full power of Contextual CNN in the more sophisticated later stages. Using a small scaling factor ($0.25$), the number of considered classes reduces rapidly at the very beginning while the features of the stage are not good enough to ensure a high recall of true labels. In practice, we decrease the scaling factor stage by stage so that the reduction of classes is slow at the beginning but gets faster and faster in the later stages. This strategy gives rise to the best result of Contextual CNN. Details of choosing the scaling factor (i.e., the number of considered classes ) for each stage are in $\S2.1$ of the supplementary material.
>
>
> |  Settings  |  Top-1 Accuracy  |
> | --- | --- |
> | Baseline ResNet50 | 76.58% |
> | factor=0.50 ($1000 \rightarrow 500 \rightarrow 250 \rightarrow 125 \rightarrow 1$) | 78.84% |
> | factor=0.25 ($1000 \rightarrow 250 \rightarrow 63 \rightarrow 16 \rightarrow 1$) | 78.51% |
> | Ours ($1000 \rightarrow 500 \rightarrow 200 \rightarrow 50 \rightarrow 1$) | **79.35%** |
>
> **Q3**: "In my understanding, in the contribution and title I would put more emphasis on the fact that the following stage of convolution depend on the results of the previous, thus rendering convolutions dynamic in a smart way."
>
> **A3**: Following your valuable suggestion, we have highlighted this point in section 1 of the revised paper.
>
>
> **A minor reminder**: we notice that you kindly acknowledged the soundness of our method with the sentences such as "The paper is simple and clear. I do not see any issues here.", but rate "Correctness" as 1. We would sincerely appreciate it if you could double-check the rating.

---

### Official Review · Reviewer_y6tk · 2022-11-02

**Confidence:** 3
**Correctness:** 4
**Technical Novelty And Significance:** 3
**Empirical Novelty And Significance:** 3
**Recommendation:** 6

**Clarity, Quality, Novelty And Reproducibility:**

This manuscript contains some intriguing ideas, one of which is this. On the other hand, the paper is difficult to follow, and the primary concept that underpins the approach that is being suggested is not simple to grasp.

**Strength And Weaknesses:**

Strength
modified Convolutional Neural Network, named Contextual Convolutional Network
deviate from this classic paradigm and propose to augment potential category memberships as contextual priors in the convolution for contextualized representation learning
trained end-to-end by standard back-propagation without additional supervision

Weaknesses
The method is not very novel and can not be regarded as a new CNN,
It is a modification of a current method that provides better results in same cases
the method is not well presented and it can be included more details about architecture
More explanations on the experiments and the discussion on results
No discussion about computation and the architecture performance
No comparison with other CNN(classic and dynamic as their definition in paper) for performance


**Summary Of The Paper:**

This paper presents a new Convolutional Neural Network, Contextual Convolutional Network, for visual recognition. Most existing convolutional backbones follow the representation-toclassification paradigm, where input representations are first generated by category-agnostic convolutional operations and then fed into perceptual classifiers (e.g., classification and segmentation).
Top-k likely classes are encoded as a contextual prior vector. Based on this vector and the previous features, convolution offsets and kernel weights are generated. The new convolutions can easily replace their plain counterparts in existing CNNs and be trained end-to-end without supervision.


**Summary Of The Review:**

The paper presented a modified architecture for CNN and it can be improved by doing some changes on presentation and adding more explanation on architecture and experiments to make the work more mature!

---

> ### Author Response · Authors · 2022-11-16
> **Response to Reviewer y6tk**
>
> We sincerely thank you for your time devoted to reviewing this paper and your constructive comments. Please see our detailed responses to your comments and suggestions below.
>
> **Q1**: "The method is not very novel and can not be regarded as a new CNN. It is a modification of a current method that provides better results in same cases."
>
> **A1**: Existing classic CNNs, e.g., ResNets (He et al., 2016), adopt the same architectures and parameters on different input features during inference. Existing dynamic CNNs, e.g., Deformable CNNs (Dai et al., 2017), adapt their structures or parameters to the input features during inference. Neither of them leverages the category priors (top-k likely classes) or modulates the convolutions to extract category-specific features. In contrast to these existing works, the main **novelty** of this work lies in modulating convolutional kernels (via weight/sampling offsets) according to the aforementioned category priors. This allows the convolutions to take on selectivity for more discriminative features w.r.t. a few most likely classes. As mentioned by Reviewer **oBzi**, our work is the first work that reveals the advantages of such practice. As mentioned by Reviewer **2om9**, our analysis is convincing and our idea is interesting and worth sharing with the community. We would be grateful if you can confirm whether our response has addressed your concerns.
>
> **Q2**: "The method is not well presented and it can be included more details about architecture."
>
> **A2**: Please refer to $\S1$ and Table 1 of the supplementary material for the detailed architecture specifications of Contextual ConvNeXt-T, Contextual ResNet50 and their plain counterparts.
> We have also highlighted this in section 3 of the revised paper.
>
> **Q3**: "More explanations on the experiments and the discussion on results." & "No discussion about computation and the architecture performance."
>
> **A3**: Following your helpful suggestions, we include more discussions of the experimental results and the computation in section 4.1 of the revised paper: "Inspired by Swin Transformer, to compare efficiency with hardware-optimized classic CNNs, we adopt an efficient batch computation approach for contextual convolutions (detailed in $\S3$ of the supplementary material). Thus, in addition to the noticeably better performances, Contextual ConvNeXt also enjoys high inference throughput comparable to ConvNeXt."
>
> **Q4**: "No comparison with other CNN (classic and dynamic as their definition in paper) for performance."
>
> **A4**: In Table 4 of the main paper, the ablation model $a1$ is a classic CNN (ResNet50). The ablation model $a3$ is a dynamic CNN that is built with the proposed contextual convolutions (w/o leveraging category priors). The ablation model $a4$ is another dynamic CNN that is built with other form of dynamic convolutions (deformable convolutions V2). Compared to all these classic/dynamic CNNs, the proposed method ($a5$) achieves significantly better performances ($+2.77\%$/$+1.76\%$/$+2.11\%$). We have further clarified this in section 4.3 of the revised paper.
>
> We hope that our responses could address your concerns. Please let us know if you have further questions.

---

### Public Comment · ~David_W._Romero1 · 2022-11-05
**Wrt global conv kernels and learnable receptive fields**

Dear authors,

thank you very much for your interesting paper!

I would like to point you to some related work which constructs and learns CNNs with global convolutional kernels [1-6]. Among these, I believe FlexConv is of particular relevance, as similarly to your approach, it learns the sizes of the convolutional kernels directly from data during training. I believe that considering FlexConv on your related work would give the reader a better idea regarding the current stand point of your promising and interesting research.

Best of luck with your submission! :)

Cheers,

 David


[1] Romero, David W., et al. "Ckconv: Continuous kernel convolution for sequential data." arXiv preprint arXiv:2102.02611 (2021).

[2] Romero, David W., et al. "Flexconv: Continuous kernel convolutions with differentiable kernel sizes." arXiv preprint arXiv:2110.08059 (2021).

[3] Romero, David W., et al. "Towards a General Purpose CNN for Long Range Dependencies in  D." arXiv preprint arXiv:2206.03398 (2022).

[4] Ding, Xiaohan, et al. "Scaling up your kernels to 31x31: Revisiting large kernel design in cnns." Proceedings of the IEEE/CVF Conference on Computer Vision and Pattern Recognition. 2022.

[5] Gu, Albert, Karan Goel, and Christopher Ré. "Efficiently modeling long sequences with structured state spaces." arXiv preprint arXiv:2111.00396 (2021).

[6] Nguyen, Eric, et al. "S4ND: Modeling Images and Videos as Multidimensional Signals Using State Spaces." arXiv preprint arXiv:2210.06583 (2022).

---

> ### Author Response · Authors · 2022-11-17
> **Response to David W. Romero**
>
> Hi David,
>
> Thanks very much for drawing our attention to the interesting prior works. In our camera ready version, we will cite the relevant papers and discuss them in the related works.

---

### Author Response · Authors · 2022-11-16
**General Response**

We sincerely thank all reviewers for the valuable time devoted to reviewing this paper and the helpful comments. The paper and the supplementary material are revised following the suggestions from all the reviewers. The revisions are in blue color.

On the one hand, we have revised **the paper** to improve its clarity and readability:
- In section 1, we highlight that "the following stage of convolutions is conditioned on the results of the previous, thus rendering convolutions dynamic in a smart way." (**oBzi**).
- In section 2, we cite and discuss SENet (**CrqV**).
- In section 3, we refer the readers to the supplementary material for detailed architecture specifications of the proposed method (**y6tk**).
- In section 3.2, we highlight that the merging layers are different between stages while the class embeddings are constant in the whole network (**2om9**).
- In section 4.1, we add more explanations/discussions on the results and the computation (**y6tk**).
- In section 4.2, we clarify that the ablation model $a1$ is a classic CNN and $a3$ and $a4$ are dynamic CNNs (**y6tk**).

On the other hand, we have revised **the supplementary material** to provide additional useful analyses:
- In $\S4$, we present the linear probe results (by fine-tuning only the top layer) of the proposed method on two fine-grained downstream datasets (**2om9**).
- In $\S6$, we discuss the failure cases of the proposed method (**CrqV**).

---

### Decision · Program_Chairs · 2023-01-20

**Decision:**

Accept: poster

**Justification For Why Not Higher Score:**

The ideas are not completely new

**Justification For Why Not Lower Score:**

The ideas are interesting and intuitive with good results. Useful to share with ICLR community

**Metareview: Summary, Strengths And Weaknesses:**

The paper presents clearly an intuitive idea of encoding top-k likely classes from the preceding stage in a network as a contextual prior vector and achieves better results over ConvNeXt-Tiny on top-1 accuracy of ImageNet classification.

**Note From Pc:**

if the above contains the word "oral" or "spotlight" please see: "oral" presentation means -> notable-top-5% and "spotlight" means -> notable-top-25%. As stated in our emails, we are disassociating presentation type from AC recommendations

**Summary Of Ac-Reviewer Meeting:**

Not needed as all reviewers agreed to accept